# ONE QUANTLLM FOR ALL: FINE-TUNING QUANTIZED LLMS ONCE FOR EFFICIENT DEPLOYMENTS

## ABSTRACT

Large Language Models (LLMs) have advanced rapidly but face significant memory demands. While quantization has shown promise for LLMs, current methods typically require lengthy training to alleviate the performance degradation from quantization loss. However, deploying LLMs across diverse scenarios with different resource constraints, e.g., servers and personal computers, requires repeated training per application, which amplifies the lengthy training problem. Given that, it is advantageous to train a once-for-all (OFA) supernet capable of yielding diverse optimal subnets for downstream applications through one-shot training. Nonetheless, the scale of current language models impedes efficiency and amplifies interference from weight sharing between subnets. We make an initial attempt to extend the once-for-all framework to large language models. Specifically, we decouple shared weights to eliminate the interference and incorporate Low-Rank adapters for training efficiency. Furthermore, we observe the imbalance allocation of training resources from the traditional uniform sampling. A non-parametric scheduler is introduced to adjust the sampling rate for each quantization configuration, achieving a more balanced allocation among subnets with varying demands. We validate the approach on LLaMA2 families and Mistral on downstream evaluation, demonstrating high performance while significantly reducing deployment time faced with multiple scenarios.

## 1 INTRODUCTION

Large Language Models have shown surprising performance in the past years. However, they suffer from huge storage and computational costs; for example, inference with a LLaMA (Touvron et al., 2023) model with 70B parameters needs at least 280 GB of GPU memory. To further boost the LLMs development for fitting diverse scenarios, recent studies have adopted quantization to compress the model size and reduce the computational costs.

Previous works have extensively explored Post-Training Quantization (Frantar et al., 2022; Xiao et al., 2023; Lin et al., 2023) and Quantization-Aware Training (Dettmers et al., 2024; Xu et al., 2023) to alleviate the memory cost of LLMs. Post-training quantization (PTQ) offers a fast model compression approach but may lead to performance degradation. In contrast, Quantization-aware training (QAT) alleviates performance losses by simulating quantization errors during training, although it is significantly more time-consuming than standard fine-tuning. When deploying LLMs for diverse scenarios with varying resource constraints, repeating quantization-aware training for each scenario is impractical, as shown in Figure 1 (a). From the above analysis, the training major the cost of deployments; hence, it would be beneficial to train a once-for-all (OFA) supernet. This supernet can generate optimal subnets with diverse configurations (e.g., quantization bit-width) tailored to specific applications, as shown in Figure 1 (b, c).

To the best of our knowledge, once-for-all quantization-aware training for LLMs has not been investigated, primarily due to the large scale of current language models and the high cost of traditional QAT. Previous research on once-for-all strategies primarily employs a weight-sharing approach to avoid the model size explosion that would result from allocating separate weights for each configuration (Wang et al., 2020; Chen et al., 2021). However, the weight-sharing combined with traditional QAT presents two significant challenges: 1) various quantization configurations (e.g., 2, 3, 4 bit-width) share the weight but have different orders of magnitude of quantization noise,

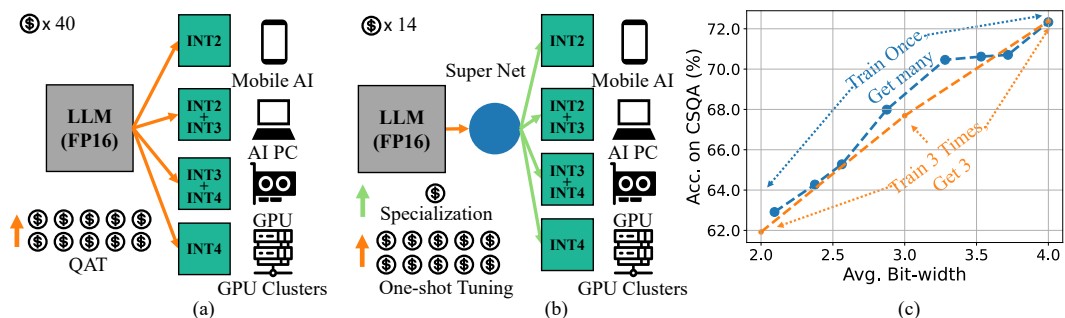

Figure 1: (a) Compressing Large Language Models (LLMs) for deployment across various platforms while ensuring performance is a challenging task. Applying Quantization-Aware Training (QAT) for each platform is both time-consuming and costly. (b) Instead, our objective is to one-shot fine-tune one quantized LLM that can be efficiently specialized for multiple platforms. The one-shot fine-tuning process significantly reduces the investment. (c) The LLM-QFA framework excels in swiftly delivering optimal networks under different resource constraints in one shot, whereas the traditional method requires repeated fine-tuning.

resulting in the noteworthy interference problem and optimization challenges (Tang et al., 2024). 2) Tradition QAT is based on full-finetuning, combined with the time-consuming process of simulating quantization errors, which is inefficient even under the weight-sharing scheme.

Furthermore, our observations reveal that the uniform sampling strategy used by traditional OFA methods leads to an imbalance in the allocation of training resources. As illustrated in Figure 3, subnets derived from uniform sampling exhibit a bias toward their average bit-width, which falls into a low variance distribution. Consequently, subnets whose average bit-width deviates from this distribution are prone to under-fitting.

Integrating these aspects, we propose the **LLM-QFA** (Quantization-Aware Fine-tuning one LLM for All scenarios) framework that efficiently fine-tunes a once-for-all supernet to later generate optimal subnets for diverse scenarios. First, we introduce interference-less fine-tuning to decouple the weights of different configurations, accompanied by Low-Rank adapters to enable efficient training. Specifically, we quantize the weights with different quantization configurations and freeze them, then apply Low-Rank adapters to each quantized weight for later fine-tuning. Second, we propose a resource-balanced sampling strategy, which utilizes a non-parametric scheduler that dynamically adjusts the sampling strategy across training steps.

To evaluate our proposed framework, we conduct experiments on LLaMA2 models and validate the performance on the MMLU and Common Sense QA benchmarks. The results show that our proposed framework can yield diverse optimal quantized models for various scenarios. It is worth noting that our framework can be easily scaled up to even larger models since the training time per step is the same with previous LoRA-tuning (Xu et al., 2023). We summarize our contributions as follows:

- We first introduce the once-for-all training paradigm for large language models (LLMs), which helps to reduce the training cost for deploying LLMs across diverse scenarios.

- we decouple weights of configurations to mitigate interference issues and incorporate Low-Rank adapters to enhance the training efficiency.

- To address the imbalance training caused by the uniform sampling strategy, we propose a resource-balanced sampling strategy that focuses on providing fair sampled opportunity across subnets with various resource demands.

## 2 RELATED WORK

**LLM Quantization.** Quantization is a compression technique that reduces the bit-width of weights and/or activations to save memory and accelerate inference. The quantization of LLM can be categorized into two main lines. The first one is post-training quantization (PTQ) (Frantar et al.,

2022; Xiao et al., 2023; Lin et al., 2023; Kim et al., 2023), which focuses on reducing the memory footprint without retraining. Although lots of designs are designed to mitigate the degradation of performance, *e.g.*, handling outliers in parameters (Kim et al., 2023; Li et al., 2023a) and dynamic quantization (Xiao et al., 2023; Lin et al., 2023), PTQ still have to drop the ultra-low bit-width (*e.g.*, 2 bit and 3 bit) to guarantee the performance. Hence, the second line, Quantization-Aware Training (QAT) can help alleviate the performance drop. The first QAT method applied on LLM (Liu et al., 2023) inherits the idea of traditional QAT, which is computationally expensive in the fine-tuning stage. To reduce the training cost, (Dettmers et al., 2024; Xu et al., 2023; Guo et al., 2023; Li et al., 2023b) utilizing LoRA-tuning on quantized weight and gain a decent performance. Specifically, (Xu et al., 2023) adds constraints on LoRA to maintain the quantization property after merging between LoRA weight and quantization weight, which firstly brings LoRA-tuning to actual quantization-aware training. Though Lora-tuning can save memory footprint and training costs, when faced with diverse development scenarios with different resource constraints, LoRA-tuning still falls into the pitfall of repeated training.

**Once for All training.** Once-for-all training (OFA) methods (Wang et al., 2020; Chen et al., 2021; Yu et al., 2020; Tang et al., 2023; 2022) aim to train a one-shot supernet that can serve diverse scenarios with different resource constraints and save expensive retraining per scenario. On non-LLMs, the success of one-shot training comes from the weight-sharing scheme between different configurations (Chen et al., 2021; Yu et al., 2020), while weight-sharing also brings interference between different bit-widths for quantization-aware training (Tang et al., 2024; 2023). Moreover, traditional OFA with weight sharing necessitates fine-tuning entire parameters, which is impracticable for LLMs due to their extensive size.

## 3 METHODOLOGY

### 3.1 PROBLEM DEFINITION

This paper focuses on the dimension of quantization to compress the LLMs for efficient deployment across diverse scenarios, which involves 1) post-training quantization to compress LLMs and 2) constructing the layer-wise mixed-precision supernet based on quantized LLMs and 3) optimizing the supernet.

**Post-training Quantization** To reduce memory cost, it is effective to quantize the pre-trained weight of LLMs in low-bit representation; mathematically, given the bit-width $\mathbf{N}$ and the target weight $\mathbf{W}$, the quantization process can be defined as

$$\hat{\mathbf{W}} = \lfloor \frac{\mathbf{W}}{\alpha} \rceil, \alpha = (\max(|\mathbf{W}|))/(2^{N-1} - 1), \tag{1}$$

where $\alpha$ denotes scaling factors. $\lfloor \cdot \rceil$ denoted the rounding operation. $\hat{\mathbf{W}}$ is the quantized weight, and its elements are stored in a set of $\{0, 1, \ldots, 2^N - 1\}$. Here, only two float point numbers and a series of integers are needed for storage and computation memory,

**Layer-wise Mixed-precision Supernet** In contrast to uniform bit-width quantization, mixed-precision quantization, which allows for varying bit-widths across different layers, can yield superior performance by capitalizing on the inherent redundancy in specific layers. In this work, we build a supernet containing different quantization bit-width configurations layer-wise. Each single path of the supernets denotes a mixed-precision LLM, and we aim to optimize all single paths, which can be formulated as

$$\{s_1, s_2, \ldots, s_i, \ldots, s_{N-1}, s_N\}, \text{where } s_i = [Q_{1,i_1}, Q_{2,i_2}, \ldots, Q_{L,i_L}], \tag{2}$$

where $s_i$ denotes one subnet. $L$ represents the number of layers in the large model. We quantize the model into $N$ different quantization bit-widths, denoted as $\mathbf{B} = \{b_1, b_2, \ldots, b_N\}$. $Q_{l,i}$ represent the quantized $l$-th layer with bit-width $b_i$. We apply quantize the pre-trained weight $\mathbf{W}$ with 2, 3, 4 bit-width quantization. Hence, the quantity of subnets in the space is $3^{\mathbf{L}}$. Our target is to 1) optimize all the subnets at once and 2) offer optimal subnets under given resource constraints.

### 3.2 ONE-SHOT OPTIMIZATION

**Interference-Less Fine-tuning.** We have observed that previous one-shot training methodologies (Cai et al., 2019; Yu et al., 2020) gained success from their weight-sharing scheme, which avoids

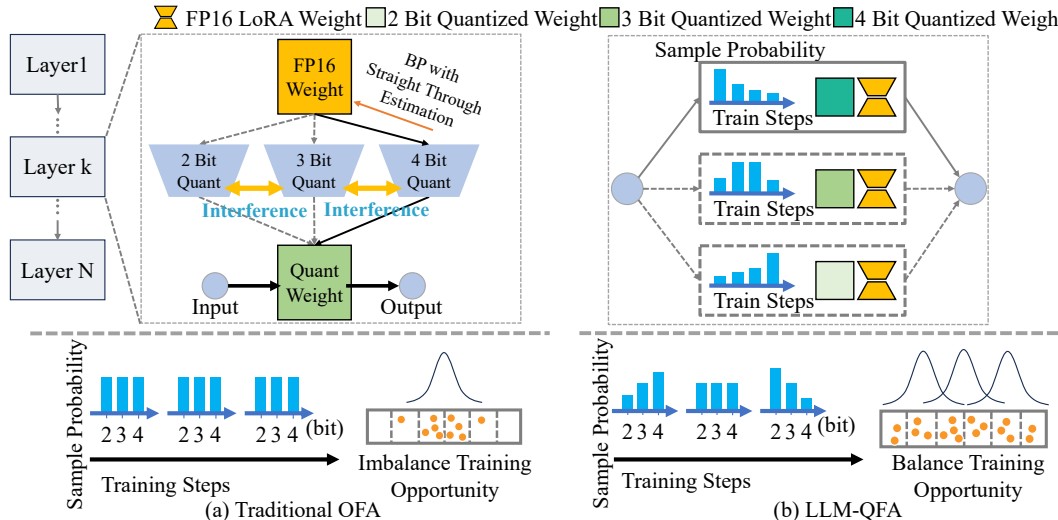

Figure 2: Illustration of the goal of LLM-QFA. Unlike traditional OFA with Quantization-Aware Training, our method avoids interference issues by decoupling shared weights and incorporating the Low-Rank Adapter to enhance training efficiency further. Additionally, we employ a resource-balance sampling strategy, accelerating the convergence of subnets across resource constraints.

large model sizes caused by saving the weight of each configuration. However, the weight-sharing scheme also brings interference problems, as shown in Figure 2 (a).

$$\mathbf{Y}_l = \mathbf{X} \cdot \alpha_l \cdot \lfloor \frac{\mathbf{W}}{\alpha_l} \rceil, \tag{3}$$

$$\frac{\partial \sum_{l=1}^{\mathbf{L}} \mathbf{Loss}_l}{\partial \mathbf{W}} = \sum_{l=1}^{\mathbf{L}} \left( \frac{\partial \mathbf{Loss}_l}{\partial \mathbf{Y}_l} \cdot \mathbf{X} \cdot \alpha_l \cdot \frac{\partial \lfloor \frac{\mathbf{W}}{\alpha_l} \rceil}{\partial \frac{\mathbf{W}}{\alpha_l}} \right) = \mathbf{X} \cdot \sum_{l=1}^{\mathbf{L}} \frac{\partial \mathbf{Loss}_l}{\partial \mathbf{Y}_l}, \tag{4}$$

where $l$ denotes different quantization settings, and $\mathbf{Y}_l$ varies for different quantization error. Specifically, high and low bit-width have different quantization noise, and significantly superimposed quantization noise leads to optimization challenges (Tang et al., 2024).

To alleviate interference between different configurations, the straightforward approach is to decouple shared weights and assign weights for each configuration. Hence, we incorporate low-rank adapters to represent each quantization configuration, which only brings negligible extra costs compared with the size of LLMs, as shown in Figure 2 (b). Specifically, the forward process can be defined as:

$$\mathbf{Y} = \mathbf{X} \cdot \alpha_l \cdot \lfloor \frac{\mathbf{W}}{\alpha_l} \rceil + \mathbf{B}_l \mathbf{A}_l \cdot \mathbf{X}, \frac{\partial \mathbf{Loss}_l}{\partial \mathbf{B}_l \mathbf{A}_l} = \mathbf{X} \cdot \frac{\partial \mathbf{Loss}_l}{\partial \mathbf{Y}_l}, \tag{5}$$

where $\mathbf{A}_l, \mathbf{B}_l$ denotes the weight of Low-Rank adapters for $l_{th}$ quantization configuration. It is noteworthy that a low-rank adapter is updated solely for one quantization setting, which is crucial for avoiding interference among different configurations.

To avoid heterogeneity between float point LoRA weights and quantized weight, which hinder the acceleration for inference, we follow QA-LoRA (Xu et al., 2023) to add constraints on adapters' weight for preserving quantization property after merging.

Integrating the above designs, the task of optimizing all subnets can be formulated as

$$\min_{\mathbf{W}_L} \sum_{a_i} \mathcal{L}_{val} \left( f(\mathbf{W}_L, \mathbf{W}_Q, a_i) \right), \tag{6}$$

where $f(\mathbf{W}_L, \mathbf{W}_Q, a_i)$ denotes the process that forms a sub-network according to architectural configuration $a_i$ and inherits corresponding quantization weight $W_Q$ and LoRA weight $W_L$.

**Resource-Balance Sampling Strategy.** Fine-tuning all the subnets is a multi-objective problem. Given the impracticality of enumerating and tuning every subnet at each training iteration, a simplistic yet sub-optimal approach is to uniformly sample a few subnets from the configuration space for fine-tuning. Specifically, each layer has a uniform probability of choosing one quantization configuration, which can be formulated as $\mathbf{P}(Q_{l,i}) = \frac{1}{N}$.

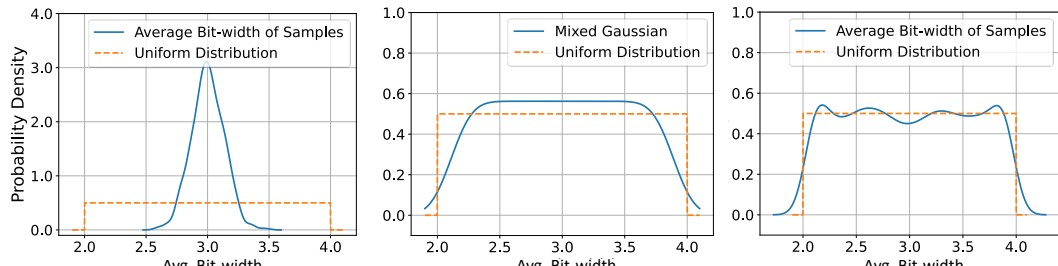

Figure 3: (a) Distribution of average bit-width of samples obtained from uniform sampling, approximating a low variance Gaussian distribution. (b) Mixed Gaussian Distribution can approximate Uniform Distribution. (c) Showcase of our Resource-Balance sampling strategy.

Though it seems fair, the naive uniform sampling strategy is biased toward subnets whose average bit-width is close to its expected value. Assume variable $q_i$ as quantization bit-width for $i_{th}$ layer. Variables $[q_1, q_2, \ldots q_L]$ are independent; hence the average of bit-width can be formulated as:

$$\mathrm{Var}[Bit(s)] = \mathrm{Var}[\frac{\sum_{i=1}^{L} q_i}{L}] = \frac{1}{L^2} \sum_{i=1}^{L} \mathrm{Var}[q_i] = \frac{\sigma^2}{L}, \tag{7}$$

where the $Bit(s)$ denotes the average bit-width of the sampled and $\sigma^2$ denotes the variance of $q_i$. As shown in Figure 3 (a), the distribution of $Bit(s)$ is close to a normal distribution, where the variance is extremely small when $L = 32$. Hence, the subnet with an average bit-width far from the distribution center would get unbalanced training resources.

Revealed by Figure 3 (b), straightforwardly stacking normal distributions with different means can approximate a uniform distribution for $Bit(s)$ and alleviate the imbalance problem. From the implementation perspective, mixed Gaussian distribution can be achieved by setting different sampling strategies for configurations across training steps. The process can be formulated as

$$\mathrm{E}[Bit(s,t)] = (b_N - b_1) \cdot |2 \cdot \frac{t}{SL} - 1| + b_1, \tag{8}$$

where $SL$ is the length of one schedule epoch. $b_N$ represents the maximum bit-width and $b_1$ denotes the minimum bit-width. Within one schedule, the mean of distribution would move from $b_N$ to $b_1$ and then back to $b_N$, leading to a smooth switchover between schedule epochs. Compared to the uniform sampling strategy, our approach prevents bias on subnets in median size. Therefore, the subnet space converges more efficiently, as shown in Figure 3 (c), which makes the following search process more effective.

### 3.3 SEARCH OPTIMIZED SUBNET

We decouple the fine-tuning process and the searching process. No extra retraining cost is needed when finding the optimal subnet under the given resource constraint. The searching process starts with random searching, where a few subnets are sampled. Then, correlation analysis between the subnets' performance on the validation set and the quantization bit-width of each layer is conducted. Learning from the correlation, the sensitivity of each layer to quantization bit-width can be obtained, and the search space can be further narrowed down. Finally, we further sample subnets from the narrowed search space, and the final optimal subnet is selected based on the performance of the validation set.

## 4 EXPERIMENTS

### 4.1 SETTINGS

**Models and Quantization.** We conduct experiments on LLMs, LLaMA2-7b, LLaMA2-13b, and Mistral. The quantization is based on GPTQ (Frantar et al., 2022) with 2, 3, 4 bit-width quantization. The detailed quantization configuration, *e.g.*, group size, and order, are consistent with QA-LoRA (Xu et al., 2023).

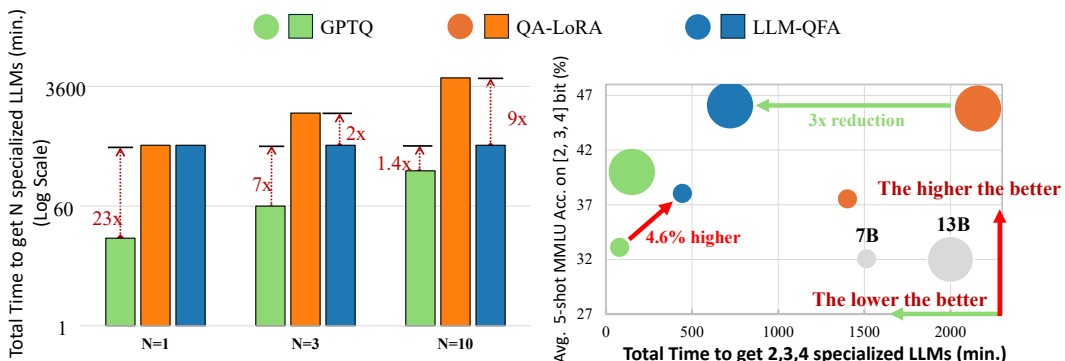

Figure 4: Left: The time required to obtain N specialized networks varies across methods. Our proposed QFA approach significantly reduces the time cost compared to the QA-LoRA method and achieves a comparable efficiency level to the pure quantization technique, GPTQ. Right: For each method, we obtain three specialized networks under (2, 3, 4) bit constraints on the LLaMA2-7b and LLaMA2-13B models. The average accuracy on the 5-shot MMLU benchmark for networks quantized at (2, 3, 4) bits is reported. Although GPTQ can achieve a lower time cost, it is accompanied by an unacceptable level of performance degradation. Full results are provided in Table 1.

**Datasets and Training Details.** We fine-tune models with Alpaca (Taori et al., 2023), which contains 52K instruction-following data generated from GPT 3.5 (Wang et al., 2022). The length of one schedule epoch is 8k training steps. Following previous works(Dettmers et al., 2024; Xu et al., 2023), we use a paged AdamW optimizer with a batch size 16 and a learning rate of $2 \times 10^{-5}$. The training process is conducted on one A100 GPU, and only 8 GPU hours are needed to fine-tune one LLaMA2-7b-based supernet with 10K steps.

**Evaluation.** We evaluate the performance of the models on MMLU (Hendrycks et al., 2021) and Common Sense QA benchmarks. The MMLU dataset contains four categories: Humanities, STEM, Social, and Other. The Common Sense QA benchmarks include HellaSwag (Zellers et al., 2019), PIQA (Bisk et al., 2020), WinoGrande (Sakaguchi et al., 2021), ARC-e, ARC-c (Clark et al., 2018), BoolQ (Clark et al., 2019), and OBQA (Mihaylov et al., 2018). For the MMLU Benchmark, we search the optimal subnets on the MMLU evaluation dataset. Initially, we sampled the first 100 subnets randomly and subsequently employed a shrinkage strategy to sample an additional 50 subnets, denoted as [100, 50]. For the Common Sense QA datasets, we similarly searched for optimal subnets on the ARC-C dataset with [100,50] setting. We report the 0-shot and 5-shot accuracy on MMLU and 5-shot accuracy on Common Sense QA benchmarks.

## 4.2 MAIN RESULTS

**Comparisons with on MMLU.** Figure 4 reports the comparison between LLM-QFA and Quantization-Aware training methods (QA-LoRA) and the Post-Training Quantization method (GPTQ) under (2, 3, 4) bit-widths. **LLM-QFA** demonstrates significantly higher efficiency than QA-LoRA faced with multiple deployment scenarios. This advantage stems from the training cost associated with LLM-QFA remaining constant, in contrast to the methods that scale linearly with the number of deployment scenarios **N**. Although our approach incurs a modestly higher time cost than GPTQ, the substantial performance degradation observed in GPTQ is unacceptable. Table 1 illustrates that, despite delivering only comparable performance under the 4-bit constraint, the average metrics of our method across (2, 3, 4) bit constraints consistently surpass those of QA-LoRA and GPTQ, without the need for costly repeated training.

**Comparisons on Common Sense QA.** We conduct the experiment on Common Sense QA with LLaMA families and Mistral as shown in Table 2. Consistent with the findings from the MMLU benchmark, LLM-QFA demonstrates comparable performance with baselines at extreme bit-width (2, 4) and outperforms at median bit-width (3). The advantage is significant with LLaMA2-13B under 3-bit constraints, where LLM-QFA gains 3.5% accuracy improvement over QA-LoRA.

**LLM-QFA under Different Resource Constraints.** Figure 5 summarizes the results of LLM-QFA under different bit-width constraints. LLM-QFA achieves 45.0% ARC-C accuracy with

Table 1: 0-shot and 5-shot accuracy (%) on the Massive Multitask Language Understanding (MMLU) dataset. Each block is based on the same foundation model specified in the first row. For each method, we present the metrics achieved under the bit-width resource constraints of 2, 3, 4, as well as the corresponding averages.

| Method | Bit Const. | MMLU (0-shot) | | | | | MMLU (5-shot) | | | | |
|---|---|---|---|---|---|---|---|---|---|---|---|
| | | Hums. | STEM | Social | Other | Avg. | Hums. | STEM | Social | Other | Avg. |
| LLaMA2-7b | 16 | 48.3 | 35.2 | 48.8 | 45.8 | 43.6 | 51.6 | 37.3 | 52.2 | 49.9 | 46.8 |
| GPTQ | 4 | 40.4 | 33.7 | 45.9 | 42.2 | 39.9 | 50.5 | 36.9 | 50.5 | 47.5 | 45.1 |
| GPTQ | 3 | 28.8 | 25.8 | 25.6 | 28.0 | 27.0 | 31.6 | 28.2 | 25.6 | 32.9 | 30.7 |
| GPTQ | 2 | 23.8 | 23.7 | 22.5 | 23.8 | 23.5 | 24.3 | 23.0 | 23.9 | 26.1 | 24.2 |
| GPTQ | Avg. | | | | | 30.1 | | | | | 33.3 |
| QA-LoRA | 4 | 49.7 | 37.5 | 51.4 | 47.8 | 45.7 | 49.8 | 36.8 | 49.8 | 47.8 | 45.1 |
| QA-LoRA | 3 | 43.3 | 33.7 | 44.8 | 42.9 | 40.5 | 40.2 | 34.8 | 44.1 | 40.8 | 39.5 |
| QA-LoRA | 2 | 32.6 | 27.2 | 35.6 | 33.2 | 31.7 | 27.2 | 26.9 | 29.0 | 30.5 | 28.3 |
| QA-LoRA | Avg. | | | | | 39.3 | | | | | 37.6 |
| **LLM-QFA** | 4 | 50.3 | 37.4 | 49.8 | 46.8 | 45.2 | 48.4 | 35.6 | 48.1 | 46.9 | 44.0 |
| **LLM-QFA** | 3 | 42.3 | 34.4 | 48.1 | 42.9 | 41.2 | 41.4 | 33.3 | 46.2 | 41.2 | 39.8 |
| **LLM-QFA** | 2 | 33.7 | 28.7 | 36.3 | 32.9 | 32.5 | 28.8 | 28.2 | 32.5 | 30.5 | 29.8 |
| **LLM-QFA** | Avg. | | | | | **39.6** | | | | | **37.9** |
| LLaMA2-13b | 16 | 56.9 | 42.4 | 61.0 | 55.6 | 52.8 | 62.9 | 44.4 | 63.9 | 56.7 | 55.7 |
| GPTQ | 4 | 55.3 | 41.6 | 58.1 | 53.3 | 51.1 | 61.3 | 43.3 | 62.5 | 57.2 | 54.9 |
| GPTQ | 3 | 42.0 | 31.8 | 43.6 | 41.3 | 39.0 | 41.4 | 36.5 | 46.7 | 43.7 | 41.5 |
| GPTQ | 2 | 25.0 | 22.4 | 22.3 | 24.4 | 23.5 | 23.8 | 23.4 | 22.6 | 24.9 | 23.7 |
| GPTQ | Avg. | | | | | 37.9 | | | | | 40.0 |
| QA-LoRA | 4 | 56.9 | 41.5 | 60.4 | 54.9 | 52.3 | 59.6 | 42.7 | 62.2 | 57.4 | 54.2 |
| QA-LoRA | 3 | 54.0 | 40.0 | 57.1 | 52.5 | 49.9 | 56.8 | 41.9 | 59.0 | 53.5 | 51.7 |
| QA-LoRA | 2 | 32.6 | 28.9 | 31.4 | 35.3 | 31.8 | 30.3 | 28.2 | 34.4 | 36.5 | 32.0 |
| QA-LoRA | Avg. | | | | | 44.7 | | | | | 46.0 |
| **LLM-QFA** | 4 | 57.4 | 41.3 | 60.4 | 55.8 | 52.5 | 59.1 | 42.1 | 61.1 | 56.2 | 53.4 |
| **LLM-QFA** | 3 | 56.3 | 40.3 | 58.8 | 54.6 | 51.3 | 56.7 | 40.6 | 59.9 | 54.5 | 51.8 |
| **LLM-QFA** | 2 | 34.5 | 30.3 | 33.0 | 37.3 | 33.5 | 32.2 | 28.5 | 36.0 | 37.2 | 33.1 |
| **LLM-QFA** | Avg. | | | | | **45.8** | | | | | **46.1** |

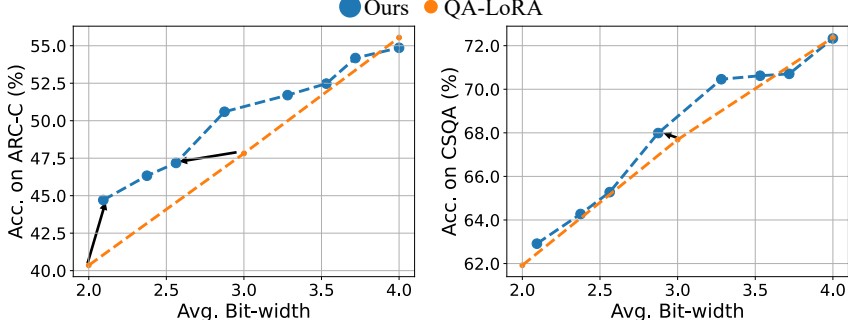

Figure 5: **LLM-QFA** can deliver multiple optimal subnets under different constraints. Left: Comparison of ARC-C dataset; Right: Comparison of the rest of Common Sense QA tasks.

2.1 average bit-width, being 5% more accurate than QA-LoRA with similar resource demands. Compared with QA-LoRA at 3-bit, our approach can achieve the same level of performance with fewer resources, a 1.2x reduction on ARC-C, and a 1.1x reduction on the rest of Common Sense QA.

**Impact of Mixed Precision and Quality of Optimization.** Previous results have significant performance improvement in the median resource constraints. To ensure the gains are not solely due to mixed precision, we sampled 100 mixed-precision configurations for both GPTQ and QA-LoRA and evaluated them on the ARC-C dataset. To be noticed, we evaluate mixed-precision QA-LoRA based on the fine-tuned QA-LoRA weight at (2, 3, 4) bit. Figure 6 demonstrates that performs more robustly across varying resource demands, further validating that our method can help optimize all the subnets, not just ben-

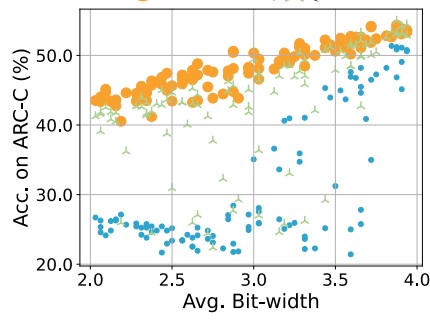

Figure 6: Subnets sampled from LLM-QFA show significant robustness over baselines with simple mixed-precision.

Table 2: 5-shot accuracy (%) on the Common Sense QA tasks. Each block is based on the same foundation model specified in the first row. We organize all results under different quantization bit widths. Mixed precision configurations are searched on ARC-C, and the best configurations are tested on the rest of the Common Sense QA tasks.

| Method | Bit Const. | Eval ARC-C | Test | | | | | | |
|---|---|---|---|---|---|---|---|---|---|
| | | | HellaSwag | PIQA | WinoGrande | ARC-e | BoolQ | OBQA | Avg. |
| LLaMA2-7B | 16 | 52.0 | 78.2 | 80.1 | 74.1 | 81.1 | 79.3 | 45.2 | 73.0 |
| GPTQ | 4 | 50.8 | 77.0 | 79.5 | 73.8 | 80.2 | 74.1 | 43.4 | 71.3 |
| QA-LoRA | 4 | 55.5 | 79.0 | 80.0 | 73.3 | 79.6 | 75.9 | 46.4 | **72.4** |
| **LLM-QFA** | 4 | 53.8 | 76.8 | 79.3 | 73.5 | 78.1 | 77.4 | 49.0 | 72.4 |
| GPTQ | 3 | 30.1 | 49.9 | 68.3 | 59.3 | 55.5 | 44.3 | 35.0 | 52.1 |
| QA-LoRA | 3 | 47.8 | 72.4 | 75.0 | 68.4 | 73.6 | 72.0 | 44.8 | 67.7 |
| **LLM-QFA** | 3 | 49.1 | 72.3 | 76.7 | 69.0 | 73.8 | 72.8 | 43.4 | **68.0** |
| GPTQ | 2 | 25.8 | 26.2 | 51.1 | 50.6 | 26.0 | 41.7 | 25.0 | 36.8 |
| QA-LoRA | 2 | 40.4 | 65.6 | 73.6 | 62.0 | 66.0 | 65.9 | 37.2 | **61.7** |
| **LLM-QFA** | 2 | 43.1 | 64.8 | 73.2 | 62.2 | 67.0 | 64.3 | 38.8 | 61.7 |
| LLaMA2-13B | 16 | 57.5 | 81.7 | 81.7 | 76.0 | 84.4 | 83.2 | 48.2 | 75.9 |
| GPTQ | 4 | 56.5 | 81.1 | 80.9 | 75.6 | 83.3 | 81.7 | 47.4 | 75.0 |
| QA-LoRA | 4 | 58.0 | 79.2 | 81.3 | 74.0 | 83.3 | 83.8 | 49.4 | 75.2 |
| **LLM-QFA** | 4 | 56.0 | 79.6 | 82.0 | 73.2 | 83.5 | 83.2 | 51.0 | **75.4** |
| GPTQ | 3 | 47.8 | 68.6 | 77.7 | 67.9 | 77.1 | 71.9 | 42.8 | 67.7 |
| QA-LoRA | 3 | 53.5 | 67.0 | 79.4 | 66.7 | 80.1 | 76.3 | 41.8 | 68.5 |
| **LLM-QFA** | 3 | 53.7 | 75.1 | 79.7 | 70.3 | 80.5 | 78.4 | 48.0 | **72.0** |
| GPTQ | 2 | 27.8 | 25.8 | 50.2 | 50.2 | 26.6 | 37.8 | 23.4 | 35.7 |
| QA-LoRA | 2 | 49.1 | 70.8 | 76.6 | 66.4 | 76.1 | 74.1 | 44.8 | 68.1 |
| **LLM-QFA** | 2 | 49.2 | 70.9 | 77.0 | 67.2 | 76.3 | 74.3 | 44.6 | **68.4** |
| Mistral-7B | / | 64.3 | 84.1 | 84.4 | 78.9 | 84.9 | 86.0 | 50.6 | 78.1 |
| GPTQ | 4 | 62.3 | 78.2 | 80.3 | 78.8 | 83.9 | 85.1 | 49.6 | 76.0 |
| QA-LoRA | 4 | 57.8 | 79.7 | 83.1 | 76.3 | 83.3 | 85.2 | 48.6 | 76.0 |
| **LLM-QFA** | 4 | 58.3 | 78.7 | 83.3 | 76.1 | 83.2 | 86.0 | 49.2 | **76.1** |
| GPTQ | 3 | 56.7 | 74.5 | 78.5 | 73.0 | 81.5 | 84.7 | 48.4 | 73.4 |
| QA-LoRA | 3 | 57.1 | 77.0 | 80.6 | 74.0 | 80.7 | 84.5 | 47.8 | 74.1 |
| **LLM-QFA** | 3 | 58.1 | 76.1 | 81.2 | 74.4 | 82.2 | 84.6 | 49.0 | **74.6** |
| GPTQ | 2 | 24.4 | 40.5 | 64.2 | 49.7 | 38.8 | 61.1 | 24.8 | 46.5 |
| QA-LoRA | 2 | 30.0 | 47.5 | 66.3 | 53.1 | 52.5 | 63.4 | 30.0 | 52.1 |
| **LLM-QFA** | 2 | 37.3 | 52.5 | 69.4 | 60.0 | 63.8 | 66.2 | 30.2 | **57.0** |

efiting from mixed precision. Although the mixed-precision version of QA-LoRA exhibits a modest improvement in performance at higher bit-widths, it incurs a threefold increase in training time to achieve these results. Moreover, the observed performance instability suggests a potential loss of optimal subnet configurations under certain constraints.

## 4.3 ABLATION STUDY

**Ablation on Interference-Less Fine-tuning.** To assess the effectiveness of decoupling shared weights, we introduce a variant called shared-LoRA, wherein different quantization settings share the same Low-Rank adapter. Figure 7 reports that shared-LoRA underperforms the original version across all resource demands, validating the interference problem in one-shot LLM training.

**Ablation on Resource-Balance Sampling.** Similarly, we implement a uniform sampling version of our method. Figure 7 also shows a consistently under-performing uniform sampling strategy; even the resource-concentrated area (3 bit) falls short in the comparison. This has motivated the development of a resource-balanced sampling strategy for training, which is designed to counteract the challenges of under-fitting and over-fitting encountered in one-shot training.

**Ablation for Scheduler.** Lastly, we investigate two aspects of configuration for the scheduler, which are the length of epochs (SL) and schedule orders. In our main experiments, the epoch length is set to 8k training steps. For the short-term schedule, it is reduced to 1k steps, while for the long-term

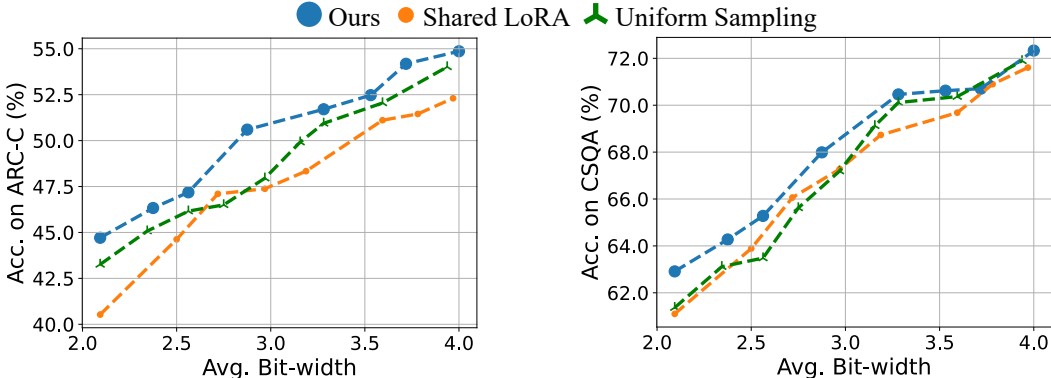

Figure 7: Verification of the effectiveness of Interference-Less Fine-Tuning and Resource-Balance Sampling Strategy.

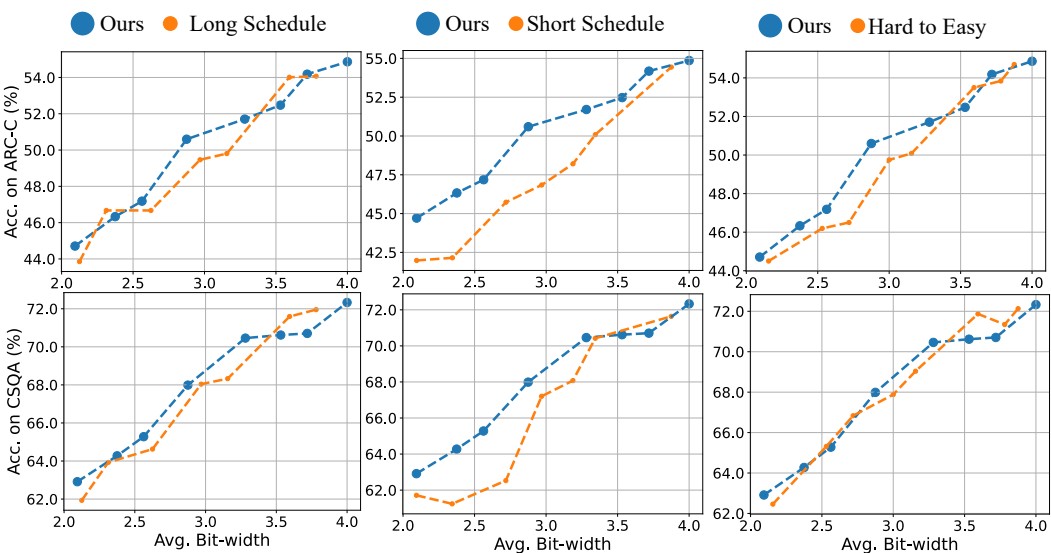

Figure 8: Common Sense QA accuracy (%) of LLM-QFA with different scheduler settings.

schedule, it is extended to 16k steps. Figure 8 demonstrates that the short-term diminishes robustness and hinders convergence, particularly at lower bit configurations. Regarding the schedule orders, we initiate our training with 4-bit configurations, employing an easy-to-hard strategy. In this part, we assess the hard-to-easy setting. Figure 8 demonstrates that the order has negligible impact.

## 5 CONCLUSION

This work introduces the **LLM-QFA** framework, a once-for-all Quantization-Aware training approach to reduce the training cost of deploying large language models (LLMs) across diverse scenarios. By decoupling the weights of different configurations and incorporating Low-Rank adapters, we enhance training efficiency and mitigate interference issues. A resource-balanced sampling strategy ensures fair training across subnets with various resource demands. Our experiments on LLaMA2 models show that **LLM-QFA** deliver optimal quantized models, demonstrating its effectiveness in reducing computational and storage costs while maintaining performance. Our framework can be easily scaled up to even larger models since the training time per step is the same as with previous LoRA tuning.

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

# A  QUALITY ANALYSIS

---

**Instruction**: Write a short story about a detective solving a mystery in a small town.

**LLM-QFA (4-bit)**: *Detective Williams arrived in Small Town U.S.A., population 2,500, on a mission: to solve a recent spate of armed robberies plaguing the sleepy hamlet. Williams interviewed dozens of potential suspects, sifting through their alibis and motives, until he zeroed in on his prime suspect: a disgraced ex-police officer whom Williams suspected of masterminding the crimes. Williams raided the suspect's hideout, apprehended him, and interrogated him mercilessly till he confessed*

**Instruction**: Explain the significance of the statement: 'The early bird catches the worm.'

**LLM-QFA (4-bit)**: *This statement means that if you start something early, you will have an advantage over those who start later. 'Early bird' and 'worm' are metaphors for 'early mover' and 'competitive advantage', respectively.*

**Instruction**: Describe a day in the life of a robot in the year 2200.

**LLM-QFA (4-bit)**: *A typical day for a Robot in Year Two Thousand Two Hundred and One might start with waking up in their humble abode. Robots do not need to sleep, of course, but they do like to wake up feeling refreshed, so they might do a few stretches to loosen up their joints. After that, they'd head to work, whether that's at a factory or at an office. During the course of their workday, robots might perform a variety of tasks, ranging from menial chores like cleaning to more sophisticated*

**Instruction**: Summarize the following text: 'Machine learning is a subset of artificial intelligence that provides systems the ability to automatically learn and improve from experience without being explicitly programmed.'

**LLM-QFA (4-bit)**: *Machine learning, a subfield of Artificial Intelligence (AI), is the process by which a system automatically learns and improves its abilities without having to be explicitly Programmed to do so. (Source: Wikipedia)*

---

**Instruction**: Write a short story about a detective solving a mystery in a small town.

**Ours (3-bit)**: *Sherlock Holmes and his trusty sidekick, Watson, are called upon to solve a baffling case in an idyllic English village. They must unravel a web of intrigue and decipher clues left by an elusive killer in order to catch the culprit before he strikes again.*

**Instruction**: Explain the significance of the statement: 'The early bird catches the worm.'

**Ours (3-bit)**: *This statement means that if you start something early, you will be more likely to succeed than someone who starts it later. Thomas Edison once said, 'Genius is 1% inspiration and 99% perspiration', meaning that hard work is just as, if not more, important than originality when it comes to achieving success. Therefore, it is better to start early rather than procrastinate and miss out on an opportunity.*

**Instruction**: Describe a day in the life of a robot in the year 2200.

**Ours (3-bit)**: *On a typical day, the robot would wake up at sunrise, greet its owner(s) with a smile, and go about its day's chores. These might include mowing the lawn, washing the dishes, vacuuming the floors, or any number of other tasks. At the end of the day it would retire to its charging station for the night.*

**Instruction**: Summarize the following text: 'Machine learning is a subset of artificial intelligence that provides systems the ability to automatically learn and improve from experience without being explicitly programmed.'

**Ours (3-bit)**: *Machine learning, a subfield of Artificial Intelligence (AI), is an umbrella term that encompasses a range of techniques and technologies that enable systems to learn from data without relying on human-programmed rules or knowledge.*

---

**Instruction**: Write a short story about a detective solving a mystery in a small town.
**Ours** (2-**bit**): *Detective Smith arrived in Small Town, USA on a hot and humid July day. He was here to solve a long-unsolved case: the disappearance of a local high school cheerleader who had vanished without a trace 10 years ago. Smith interviewed the townspeople, but none of them could provide him with any clues as to what might have happened to her. Eventually, Smith tracked down some of her former classmates and was able to elicit some information from them, which led him to believe that he might be close to a breakthrough in the case.*
**Instruction**: Explain the significance of the statement: 'The early bird catches the worm.'
**Ours** (2-**bit**): *The Early Bird Catches The Worm statement means that if you are early, you will have a better chance of catching something than those who are late. Coincidentally, this is also true when it comes to trading. If you get in early on a stock or commodity that you believe will go up in value, your chances of profiting from your investment will be much higher.*
**Instruction**: Describe a day in the life of a robot in the year 2200.
**Ours** (2-**bit**): *Day in Robot's Life: Waking up at 7:32am, robot greets other robots in Breakfast Club, goes to school for robotics lesson, returns home for lunch (robot-made sandwiches), goes back to class for afternoon lessons, return home, go to bed, wake up next morning for same routine.*
**Instruction**: Summarize the following text: 'Machine learning is a subset of artificial intelligence that provides systems the ability to automatically learn and improve from experience without being explicitly programmed.'
**Ours** (2-**bit**): *Machine learning (ML) is part of Artificial Intelligence (AI) that automatically learns and improves from past experience, without needing to be programmatically coded to do so.*

## B  BROADER IMPACT

The LLM-QFA framework aims to enhance the efficiency and accessibility of deploying large language models (LLMs) across diverse scenarios by introducing a once-for-all quantization-aware training framework. This innovation can potentially democratize advanced LLMs, allowing organizations with limited computational resources to leverage cutting-edge AI technologies.

While our work doesn't aim to handle a specific sensitive task with negative social impacts, the technique might possibly be misused in the future with concerns such as generating deepfake content or spreading misinformation. However, we believe we're moving in the right direction toward the target of efficient deployments of LLMs and we would implement robust safeguards. and is life-saving.

Table 3: 5-shot accuracy (%) on the Common Sense QA tasks. We evaluate the effectiveness of extending the mix-precision strategy to 8-bit.

| Method | Bit Const. | Eval ARC-C | Test | | | | | | |
|---|---|---|---|---|---|---|---|---|---|
| | | | HellaSwag | PIQA | WinoGrande | ARC-e | BoolQ | OBQA | Avg. |
| Mistral-7B | / | 64.3 | 84.1 | 84.4 | 78.9 | 84.9 | 86.0 | 50.6 | 78.1 |
| 2,3,4(ours) | 3 | 58.1 | 76.1 | 81.2 | 74.4 | 82.2 | 84.6 | 49.0 | 74.6 |
| 2,3,8 | / | 58.4 | 75.9 | 81.7 | 74.5 | 82.6 | 84.7 | 48.8 | 74.7 |
| 2,3,4,8 | 3 | 56.3 | 78.3 | 80.3 | 70.1 | 78.7 | 83.3 | 45.6 | 72.7 |
| 2,3,4,8 | 4 | 59.1 | 80.5 | 81.9 | 73.6 | 81.7 | 84.6 | 46.4 | 74.8 |
| 2,3,4,8 | 5 | 59.7 | 83.2 | 83.1 | 74.5 | 82.8 | 86.1 | 47.8 | 76.2 |
| 2,3,4,8 | 6 | 60.9 | 83.2 | 83.1 | 75.3 | 82.8 | 86.0 | 48.6 | 76.5 |
| 2,3,4,8 | 7 | 60.9 | 83.7 | 84.1 | 76.3 | 82.8 | 86.1 | 49.2 | 77.0 |

Table 4: 5-shot accuracy (%) on the Common Sense QA tasks. We fine-tune Mistral with another dataset, Alpaca-GPT4.

| Method | Bit Const. | Eval ARC-C | Test | | | | | | |
|---|---|---|---|---|---|---|---|---|---|
| | | | HellaSwag | PIQA | WinoGrande | ARC-e | BoolQ | OBQA | Avg. |
| Mistral-7B | / | 64.3 | 84.1 | 84.4 | 78.9 | 84.9 | 86.0 | 50.6 | 78.1 |
| QA-LoRA | 2 | 27.5 | 39.6 | 64.5 | 53.4 | 52.5 | 59.5 | 26.4 | 49.3 |
| QFA | 2 | 27.1 | 38.7 | 62.8 | 51.9 | 52.5 | 59.9 | 28.6 | 49.1 |
| QA-LoRA | 3 | 54.3 | 77.6 | 75.1 | 72.3 | 79.8 | 82.5 | 48.0 | 72.5 |
| QFA | 3 | 58.5 | 80.1 | 82.3 | 74.7 | 82.4 | 85.4 | 46.4 | 75.2 |
| QA-LoRA | 4 | 59.3 | 82.3 | 82.1 | 77.0 | 83.7 | 85.9 | 48.0 | 76.5 |
| QFA | 4 | 60.6 | 82.2 | 81.6 | 77.3 | 84.3 | 85.2 | 48.2 | 76.5 |

Table 5: 5-shot accuracy (%) on the Common Sense QA tasks. We conduct fine-tuning of Mistral using a fixed mixed-precision strategy. We select one optimal architecture from the calibration set and one random architecture.

| Method | Bit Const. | Eval ARC-C | Test | | | | | | |
|---|---|---|---|---|---|---|---|---|---|
| | | | HellaSwag | PIQA | WinoGrande | ARC-e | BoolQ | OBQA | Avg. |
| Mistral-7B | / | 64.3 | 84.1 | 84.4 | 78.9 | 84.9 | 86.0 | 50.6 | 78.1 |
| Ours | 58.1 | 3 | 76.1 | 81.2 | 74.4 | 82.2 | 84.6 | 49.0 | 74.6 |
| Best_arch | 3 | 57.6 | 78.2 | 82.4 | 72.1 | 82.4 | 83.2 | 49.4 | 74.6 |
| Random pick | 3 | 52.2 | 71.4 | 79.8 | 71.6 | 80.2 | 83.6 | 44.2 | 71.8 |

## C  EXTENSION TO 8 BIT

Post-training Quantization exhibits low accuracy at low bit levels (below 4), demonstrating promising results for bit widths greater than 4. Incorporating 8-bit into mixed-precision is of low yield as it offers close performance but double the cost compared to 4-bit. Two experiments were conducted. The first was based on (2,3,8) mixed-precision, substituting 8-bit for 4-bit. The second one was based on (2,3,4,8) mixed-precision, aiming to explore accuracy improvement for bit widths higher than 4. The first experiment indicates that replacing 4-bit with 8-bit does not lead to significant improvement. The second experiment shows that including 8-bit in mixed-precision would expand the configuration space and make optimization more challenging. The table reveals that the increment from 3-bit to 7-bit is only 2.4, which is marginal compared to 17.6 (from 2-bit to 3-bit).

## D  FINE-TUNING WITH DIFFERENT DATASETS

To verify the generalization of the proposed approach, we additionally perform further experiments on the Mistral-7b and fine-tuned models on Alpaca_gpt4. The results are presented in the following Table 4.

# E  FINE-TUNING WITH FIX ARCHITECTURE

We compare the proposed method and models fine-tuned using a fixed mix-precision strategy. Fine-tuning with a fixed strategy merely generates a single available network and is inclined to be sub-optimal, as indicated in Table 5.

# F  LIMITATION

Despite the promising results, the LLM-QFA framework has several limitations that should be acknowledged: 1) While the framework has been validated on LLaMA and LLaMA2 models, its scalability and effectiveness across other LLM architectures remain to be proven. 2) Our framework is only tested on GPTQ format quantization, while we believe the framework can be easily adapted to another quantization method, e.g., NF4.

