# OpenReview forum: "One QuantLLM for ALL: Fine-tuning Quantized LLMs Once for Efficient Deployments"
_ICLR.cc/2025/Conference — Submitted to ICLR 2025_

### Official Review · Reviewer_s7p9 · 2024-11-02

**Soundness:** 3
**Presentation:** 3
**Contribution:** 3
**Rating:** 5
**Confidence:** 4

**Summary:**

This work tackles a challenge in deploying large language models (LLMs) across multiple platforms with different computing resources without the need for repeated fine-tuning. Specifically, when targeting different deployment environments (e.g., from cloud to edge devices), varying quantization settings (such as different bit-widths) may be required, and each configuration usually demands separate fine-tuning to maintain accuracy. This process is computationally expensive. To address this challenge, the paper introduces LLM-QFA, a framework designed to streamline the deployment of LLMs across diverse hardware by performing quantization-aware fine-tuning only once, rather than retraining for each scenario.

**Strengths:**

Given the high number of potential layer-by-layer quantization configurations, an exhaustive search is impractical. To this end, the proposed method consists of a set of techniques, including:
1. Identify layers that are most sensitive to quantization, focusing the search on configurations likely to yield better performance.
2. Adjust sampling probabilities dynamically to ensure all configurations receive balanced training, reducing biases toward certain bit-widths.
3. Perform one-shot fine-tuning across configurations, then use a validation-driven search to select the best configurations without retraining.
4. Use varied bit-widths across layers, focusing higher precision where it matters most, to achieve efficient, high-performing subnets.

Overall, the proposed method demonstrates competitive or better accuracy compared to GPTQ and QA-LoRA, with less computational cost. In addition, LLM-QFA’s resource-balanced sampling strategy outperforms random and uniform sampling.

**Weaknesses:**

One concern that the reviewer has is that quantization alone may not be the most challenging problem for cloud-to-edge cross platform deployments, instead, the key challenge is the required orders-of-magnitude variance in terms of the total numbers of model parameters. Moving from cloud-level hardware to highly constrained edge devices typically requires a significant decrease in the number of model parameters, often through methods such as pruning or model distillation, and LLM-QFA does not inherently address the orders-of-magnitude compression needed to bridge the cloud-to-edge gap.

**Questions:**

Given the high number of potential layer-by-layer quantization configurations, an exhaustive search is impractical. However, it would be valuable if the authors could analyze the potential performance gap between LLM-QFA and a hypothetical exhaustive search. Such an analysis could help quantify how close LLM-QFA comes to true optimal configurations.

---

> ### Author Response · Authors · 2024-11-23
>
> **Application on cloud-to-edge cross-platform deployments**
> We appreciate your suggestions. Applying 3-bit quantization to a 7B model yields a 7*3/16=1.31B model, suitable for edge devices in terms of storage. However, the actual computation cost is larger than the original 1.3B model due to the dequantization process and the larger hidden size. However, quantization could still serve edge-side models with lower I/O costs and lower computation costs, which is also important for edge devices. Due to the limited computational resources, experiments on edge-side models are still in progress.
>
> **Exhaustive search optimal configurations**
> Good point! Rather than conducting an exhaustive search, we sample 100 configurations and obtain the corresponding downstream performance.
> Next, we utilize the Python package Fitter to identify the distribution, such as a normal distribution or a gamma distribution. Finally, we estimate the upper bound of the performance based on the distribution. For example, if the distribution is normal, we can estimate the upper bound by the mean + 3*std. The upper bound is used to determine the optimal subnet.
> In the case of LLaMA-3-8B, 100 samples of the downstream performance of the 3-bit model fit a normal distribution. Then we estimate the upper bound of the performance to be 0.42624 + 3 * 0.029930 = 0.516032, where our method achieves 0.492.

---

### Official Review · Reviewer_YayH · 2024-11-02

**Soundness:** 3
**Presentation:** 3
**Contribution:** 2
**Rating:** 6
**Confidence:** 3

**Summary:**

This paper presents LLM-QFA, a framework enabling once-for-all quantization-aware training of Large Language Models for efficient deployment across different hardware scenarios. The key contributions are weight decoupling to eliminate interference between quantization configurations and a resource-balanced sampling strategy for fair training across bit-widths. Evaluated on LLaMA2 and Mistral models, the framework achieves comparable performance to QA-LoRA and GPTQ baselines while requiring only one-time training instead of repeated fine-tuning for each quantization setting.

**Strengths:**

* The paper propose novel attempt to apply once-for-all training paradigm to LLM quantization, addressing weight sharing interference issues.
* The paper provides strong empirical results showing competitive or better performance vs baselines (QA-LoRA, GPTQ) while requiring only one-time training.
* The paper includes clear technical writing and comprehensive ablation studies validating each component.

**Weaknesses:**

* The main motivation around reducing training time overhead seems weakly justified, as quantization-aware training is typically a one-time procedure and not a time-critical task.
* The author only did limited experimental validation across model architectures (only tested on LLama2 and Mistral). What about models like Llama3 series?

**Questions:**

* Have the authors considered evaluating on other popular LLM architectures like Llama3 to demonstrate broader applicability? Also are there any results on larger models like Llama3-70B?
* Given that quantization-aware training is typically a one-time procedure per model, could the authors better justify why reducing training time is a critical problem?

---

> ### Author Response · Authors · 2024-11-23
>
> **Test on LLaMA-3-8B (W2, Q1)**
> Thanks for your suggestions. We have tested on LLaMA-3-8B, and the results are shown below. We follow the same experimental setting claimed in the paper, and the results are consistent with other models. However, limited by computational resources, we are currently unable to test on larger models. We will update the results in the revised version.
>
> | Bit   | Method  | Arc_challenge | winogrande | openbookqa | boolq  | arc_easy | Average |
> |-------|---------|---------------|------------|------------|--------|----------|---------|
> | 2 bit | QA-LoRA | 30.63         | 54.70      | 31.40      | 60.70  | 55.00    | 46.49   |
> | 2 bit | QFA     | 31.65         | 54.61      | 31.40      | 61.98  | 57.23    | 47.37   |
> | 3 bit | QA-LoRA | 48.63         | 67.80      | 44.20      | 77.90  | 74.52    | 62.61   |
> | 3 bit | QFA     | 49.23         | 70.08      | 42.20      | 80.51  | 75.88    | 63.58   |
> | 4 bit | QA-LoRA | 58.10         | 76.40      | 50.00      | 84.89  | 84.46    | 70.77   |
> | 4 bit | QFA     | 59.47         | 76.00      | 48.60      | 85.90  | 84.55    | 70.90   |
>
> **Motivation of OFA for LLMs (W1, Q2)**
> As stated in Lines 14-16, the quantization deployment of LLMs would yield varying compression rates based on specific requirements, such as reduced I/O costs and reduced energy consumption. However, low-bit quantization (2-bit, 3-bit) experiences an accuracy drop, necessitating extensive quantization-aware training. In previous settings, each time we were deploying for one scenario, we needed to retrain the model from scratch. Motivated by the aforementioned issues, we propose a framework that can simultaneously provide optimal subnets with varying compression rates, thereby efficiently serving diverse platforms with varying requirements.

---

> > ### Comment · Reviewer_YayH · 2024-11-24
> >
> > Thank you for your thoughtful rebuttal. I maintain my rating.

---

### Official Review · Reviewer_D3na · 2024-11-03

**Soundness:** 3
**Presentation:** 3
**Contribution:** 2
**Rating:** 6
**Confidence:** 4

**Summary:**

This paper argues that, in the context of training multi-width LLM OFA supernets,  decoupling shared weights to eliminate interference from weight sharing between subnets, using low-rank adapters for training efficiency, and dynamically adjusting the sampling strategy across training steps improves deployment time. The paper provides experimental evidence supporting this argument.

**Strengths:**

The paper addresses the important problem of enabling OFA training of multi-width quantized LLM supernets and allows this problem to be solved more efficiently than brute-force techniques while maintaining accuracy.

It works. It improves training time compared to QA-LoRA roughly in proportion to the number of specialized models required.

Although not extremely novel, it is a fun and interesting paper to read. The authors do a pretty good job explaining the challenges they encountered and motivating their solutions. There is one portion of the paper (on the problem with using uniform sampling) that I think could be clearer. I think conference attendees might find this interesting even if they don't need to use the specific approach the paper describes.

**Weaknesses:**

I think this is incremental work. Not very novel but useful and has some novelty. It applies concepts developed in other contexts in a new related context. The observations on the implications of uniform subnet sampling during training are interesting, but not well explained.

There are some grammatical errors, e.g., "the training major the cost of deployments".

There has been some recent work claiming that interference mitigation in OFA training isn't very helpful. The authors of the paper under review claim that this problem is important and describe a potential solution. Contrasting with prior work that makes a contrasting argument would improve the paper. I have an example paper in mind, but I do not want to push you into citing a particular paper. However, I think you might find some interesting and apparently contradictory work if you do a quick survey, and contrasting with it would improve your paper.

The paper finds that that layer-wise uniform sampling of subnets is biased toward subnets with mean widths near the mean bit width. This, if I understand correctly, is a straight-forward implication of the Central Limit Theorem. However, it is important to consider what specific thing ought to be uniformly sampled: subnets or subnet mean widths. If I understand correctly, there should be more subnets with mean widths near the global mean, so one must sample them more frequently to distribute training evenly across subnets. The paper, rather implicitly and with little discussion, implies that uniform sampling over subnet mean widths is a more desirable goal. You can make samples uniform over subnets or subnet mean widths, not both. Prior work chose "subnets". You choose "subnet mean widths". Why? Your evaluation shows that it works a little bit better. Is that sampling noise or a genuine improvement? If it's genuine, do you have (ideally testable) ideas on why your notion of uniform works better in practice?

**Questions:**

The last paragraph in the Weaknesses section of my review is in effect also a question. We can discuss that one here. I think it's more likely an opportunity to revise for clarity instead of a fundamental flaw.

---

> ### Author Response · Authors · 2024-11-23
>
> Thanks for your suggestions. We will correct the typos in the revised version.
>
> For main concern:
>
> **Interference mitigation in OFA**
> Good point! After conducting a survey, we came across a related paper titled "Does Interference Exist When Training a Once-For-All Network" [https://arxiv.org/pdf/2204.09210]. experimenting on small CNN models. As previously suggested, the interference problem becomes severe when the model size increases [1]. Hence, previous works focus on small models, e.g., MobileNetV3; ResNet would have fewer interference problems.
>
> **Subnets or Subnet Mean Widths**
> Another good catch! One of the main differences between our work and the previous OFA is that we decouple the shared weights with low-rank adapters, thereby reducing interference. However, the decoupling process results in fewer updates for each weight and server during the fitting problem. Another concern is that even after quantization-aware training, the 2-bit quantization still lags significantly behind the 4-bit quantization. As shown in Figure 7 left, the 4-bit model achieves 55% while the 2-bit model only achieves 45%. ARC-C contains single-choice questions with four options, where the random guess accuracy is 25%. From the perspective of a specific layer, the remaining subnets represent 'context'. The uniform sampling strategy centers the subnets around 3 bits, leaving the training unaware of the 2-bit 'context.' Although we have not yet tested those insights, Figure 7's result confirms the effectiveness of our method.
>
> [1] Retraining-free model quantization via one-shot weight-coupling learning[C]

---

> > ### Comment · Reviewer_D3na · 2024-11-24
> >
> > Thanks for the replies. I see that you were able to well deal with the first question. Your response to the second seems reasonable to me. It seems that dealing wtih that question thoroughly, e.g., by determining with (statistical) confidence that your proposed distribution works better, would likely take more time than is built into the review process.

---

> > > ### Author Response · Authors · 2024-11-25
> > >
> > > Thank you for acknowledging our replies solve your questions. Could you please consider increasing your rating to reflect our responses in the rebuttal? Meanwhile, we would like to discuss more with you if you have any further questions.

---

### Official Review · Reviewer_SLP8 · 2024-11-06

**Soundness:** 2
**Presentation:** 3
**Contribution:** 1
**Rating:** 1
**Confidence:** 4

**Summary:**

This paper introduces LLM-QFA, a supernet-based approach that fine-tunes multiple quantized models with identical architectures but varied bit widths to cater to different deployment scenarios. The main contribution lies in integrating quantization-aware training (QAT) with LoRA, allowing different subnets to share the same LoRA component. Comparisons are drawn against GPTQ and QA-LORA, with claims of improved accuracy over GPTQ and faster fine-tuning than QA-LORA.

**Strengths:**

The paper is well-organized and clear in its explanations, with the exception of the details surrounding equation (8), which could be further clarified.

**Weaknesses:**

**1. Practical Relevance of the Problem:**

The proposed approach addresses a problem that lacks practical application. In real-world deployments, training a large array of subnets to cover various quantization configurations is typically unnecessary. Practitioners often select a few configurations (e.g., 2-bit, 4-bit, or 8-bit quantization) rather than an exponential number, as suggested in the paper. This exponential approach not only demands excessive computational resources but also offers limited benefits. For scenarios requiring mixed-precision models, existing methods (e.g., “Hessian-Aware Quantization of Neural Networks with Mixed-Precision,” NeurIPS 2020) strategically determine which layers to quantize without needing such an extensive range of models. Thus, the paper’s proposed problem appears impractical and lacks substantial real-world applicability.

**2. Misalignment Between Proposed Approach and Evaluated Scenarios:**

The proposed approach is positioned within a Once-for-All-Training (OFA) framework, which suggests the need to train numerous models simultaneously. However, the evaluation focuses only on fine-tuning 2-bit, 3-bit, and 4-bit quantized models, with no consideration of other configurations. This discrepancy affects the validity of the reported execution time, as it excludes the cost of fine-tuning all potential bit-width combinations, thereby leading to an unfair comparison with QA-LORA. As it stands, the execution time presented does not accurately reflect the time required for the complete proposed approach, making the comparison with baselines misleading.

**3. Choice of Baselines:**

The selection of baselines lacks fairness. The proposed approach uses quantization-aware training (QAT), yet GPTQ, a post-training quantization method, is chosen as a baseline. A more appropriate baseline would employ QAT as well, providing a more equitable comparison.

**4. Low Novelty:**

The approach primarily combines existing techniques—quantization-aware training and LoRA—within an OFA context, but it does not substantively test the full implications of this setting. Without new insights or substantive contributions, the approach appears to merely integrate established methods without yielding notable theoretical or practical advancements.

**5. Ambiguity in Explanation of Equation 2:**

Equation (2) is not adequately explained, leading to potential misinterpretations. For example, the meaning of certain variables, such as “t”, remains unclear.

**6. Suspicious Results in Table 1:**

Some reported results in Table 1 raise concerns. For LLaMA2-13b, the QA-LoRA approach claims 0-shot accuracy values of 52.3%, 49.9%, and 31.8% for quantization widths 4, 3, and 2, respectively; however, their average does not match the reported 45.3%. Similarly, for another setting, it reports 5-shot accuracy values of 54.2%, 51.7%, and 32.0% for the same widths, but the average is again inconsistent with the reported 45.8%. These discrepancies suggest possible errors, raising doubts about the thoroughness of the entire experimental validation.

**Questions:**

1. Could the authors compare their approach with a baseline that also employs quantization-aware training?

2. Could they report the total execution time for training all bit-width combinations? (On page 3, the paper states that the number of subnets is L^3, where L is the number of layers.)

3. Can the authors provide further justification for the practical relevance of the addressed problem?

4. Could they clarify the novelty of the proposed approach in terms of unique contributions?

---

> ### Author Response · Authors · 2024-11-23
>
> We appreciate your insightful comments and suggestions. We will address your concerns and suggestions in the revised version.
>
> **Practical Application (W1)**
> Quantization serves LLMs in two aspects: lower I/O cost (weight-only quantization) and faster computation (weight-activation quantization). Currently, mainstream GPUs only support matrix multiplication up to 8 bits, which is the typical scenario we use for 8-bit quantization. For weight-only quantization, the 4-bit quantization is the most common choice, as it can achieve a comparable accuracy with 8-bit quantization. In other words, there is no need to apply (4,8) mixed precision quantization.
> You can integrate the weight-only quantization and weight-activation quantization technique by: 1. storing weights in low-bit format, such as 4 bits; 2. dequantizing weights to 8 bits and quantizing activations to 8 bits; and 3. performing matrix multiplication with 8-bit weights and activations. These integrations are widely used in the industry (Qserve, https://arxiv.org/abs/2405.04532), and those two techniques are orthogonal to each other.
> To further compress the model, we can apply lower bit quantization, e.g., 2 bits, 3 bits. While the lower bit quantization suffers from an accuracy drop, it requires heavy retraining to recover the accuracy.
> The main contribution of our work is to provide a framework that offers optimal subnets with different compression rates at the same time. The diverse optimal subnets can efficiently serve different platforms with different requirements.
> Moreover, it only requires the same level of training resources compared with retraining one model with traditional QAT (e.g., QA-LoRA, Q-LoRA, HAWQ).
>
> **Alignment between methods and evaluation metrics (W2)**
> To compare with **non-uniform** quantization, we have provided the evaluation and ablation study on more fine-grained bit-width, as shown in Figures 5, 6, 7, and 8. Those results validate the effectiveness of our method at serving multiple subnets with different compression rates.
>
> To compare our method with the **uniform** quantization method, we compare it with QA-LoRA at the same compression rate, as presented in Tables 1 and 2.
> Though our method fine-tunes models with all potential bit-widths, the proposed resource-balancing sampling strategy makes the process efficient, and the whole process requires the same time as QA-LoRA. For LLaMA-2-7B, it takes 12 hours to fine-tune and 0.5 hours to find the optimal subnet. As previously claimed, the framework serves to provide optimal subnets at one time, where previous works need 12*3=**36** hours and ours need 12+0.5*3=**13.5** hours. The time saving is significant, and we believe the comparison is fair.
>
>
> To ablate the effect of mixed precision quantization, we also extend QA-LoRA with mixed-precision training, as shown in the table. The results indicate that fine-tuning with a specific cherry-picked mixed precision setting does not yield any improvement over our method, which can provide more diverse subnets within the same training time. The cherry-picked subnet is selected based on its optimal performance on the validation set.
>
> | Method | ARC_c  | HellaSwag | PIQA   | WinoGrande | ARC_e  | BoolQ  | OBQA   | Avg.   |
> |---|---|-|---|--|---|---|---|---|
> | Ours   | 58.1  | 76.1| 81.2  | 74.4 | 82.2  | 84.6  | 49.0  | 74.6  |
> | Cherry Pick   | 57.6  | 78.2| 82.4  | 72.1 | 82.4  | 83.2  | 49.4  | 74.6  |
> | Random pick | 52.2  | 71.4| 79.8  | 71.6 | 80.2  | 83.6  | 44.2  | 71.8  |
>
>
> **Choice of Baselines (W3)**
> As previously claimed, pure low-bit quantization suffers from accuracy drop; hence we need QAT. GPTQ's presence demonstrates the effectiveness of QAT methods in restoring accuracy, facilitating a more thorough comparison.

---

> ### Author Response · Authors · 2024-11-23
>
> **Novelty (W4)**
> We would like to clarify that achieving OFA for LLMs is **non-trivial** due to their substantial size. As mentioned in our paper, the **interference problem** becomes severe when the model size increases [3], and, **for LLMs, it is unbearable to keep the weights of each configuration in memory**. Hence, we utilize low-rank adapters to decouple the shared weights. It is worth noting that our method **doesn't rely on specific LoRA techniques**, and any adapter-like design, e.g., PEQA [1], QLoRA [2], can meet our requirements.
>
> As mentioned in Sec. [3.2], uniform sampling leads to an over-sampling issue for certain sub-networks, which lowers the optimization chance for other sub-networks and causes a severe **underfitting** problem. Therefore, we suggest implementing a **resource-balanced sampling strategy** by adjusting the pick rate for each layer's configuration using the following equation. The downstream results, as shown in Fig. 7 and the attached PDF, show that our method can achieve better performance than uniform sampling and further validate its effectiveness. We sincerely hope you will consider our insights and contributions. **1. Solving the interference problem 2. solving the imbalance problem**, which are acknowledged by other reviewers.
>
> **Ambiguity in Equation**
> Thanks for your suggestions. 't' in equation (8) represents the training step; we would clarify it in the revised version.
>
> **Results in Table 1**
> We appreciate your findings.
> The mistake is caused by the late update of the average after rerunning 2-bit QA-LoRA with asymmetric quantization (initially we run 2-bit QA-LoRA with symmetric quantization).
> After double-checking all the experiments, we correct the mismatch, and the results are updated in the revised version.
>
> [1] Memory-efficient fine-tuning of compressed large language models via sub-4-bit integer quantization[J]
>
> [2] Qlora: Efficient finetuning of quantized llms[J]
>
> [3] Retraining-free model quantization via one-shot weight-coupling learning[C]

---

> ### Comment · Reviewer_SLP8 · 2024-12-03
>
> I have carefully read the authors' responses. I still think the proposed framework is not useful in practice due to the points that I mentioned in W1. I also think the novelty is limited. I decide to maintain my score.

---

### Meta-Review · Area_Chair_jFaF · 2024-12-20

**Metareview:**

The paper presents a methodology for Fine-tuning Quantized LLMs in a train a once-for-all (OFA) setting, that involves fine-tuning multiple quantized models with identical architectures but varied bit widths to cater to different deployment scenarios.  By decoupling shared weights to eliminate interference from weight sharing between subnets, using low-rank adapters for training efficiency, and dynamically adjusting the sampling strategy across training steps the paper improves deployment time. Experiments are conducted on LLaMA2 families and Mistral, as well as on LLaMA-3-8B at rebuttal time.

Strengths of the paper include the importance of the "quantize-for-all", approach, the use of LoRA finetuning to deal with interference, and the ideas introduced around sampling to improve speed while remaining performant in terms of accuracy.

On the negative side, ideas were deemed interesting but not particularly novel-existing techniques are stitched together. Reviewers also pointed out some additional issues that collectively indicate that the paper is not ready for prime time: numerical errors were made in presenting the results, important and quite pertinent related work was missed, and experiments could have been more extensive (involved more models and baselines), Another defect is that, even though the motivation lies in deployments in resource constrained hardware, no such deployment is actually made or evaluated; as the authors acknowledge, timing results may not be as anticipated, but other pertinent issues may also only be observed when an actual execution on resource constrained hardware occurs. The authors are encouraged to address all of the above points in a resubmission.

**Additional Comments On Reviewer Discussion:**

A reviewer indicated that the sampling policy is of interest, but specific choices behind sampling could be better motivated and explored. This is an additional future direction to pursue. Two reviewers challenged the practical advantages of QFA, but others were not as concerned; an implementation across hardware would help settle this.

---

### Decision · Program_Chairs · 2025-01-22

Reject